# Effects of Calcium and pH on Rheological Thermal Resistance of Composite Xanthan Gum and High-Methoxyl Apple Pectin Matrices Featuring Dysphagia-Friendly Consistency

**DOI:** 10.3390/foods13010090

**Published:** 2023-12-26

**Authors:** Huaiwen Yang, Liang-Yu Chou, Chi-Chung Hua

**Affiliations:** 1Department of Food Science, National Chiayi University, Chiayi City 60004, Taiwan; 2Department of Chemical Engineering, National Chung Cheng University, Chiayi City 621301, Taiwan

**Keywords:** high-methoxyl pectin, thermal processing, xanthan gum, rheological properties, texture profile analysis, dysphagia

## Abstract

High-methoxyl apple pectin (AP) derived from apple was employed as the main ingredient facilitating rheological modification features in developing dysphagia-friendly fluidized alimentary matrices. Xanthan gum (XG) was also included as a composite counterpart to modify the viscoelastic properties of the thickened system under different thermal processes. The results indicate that AP is extremely sensitive to thermal processing, and the viscosity is greatly depleted under a neutral pH level. Moreover, the inclusion of calcium ions echoed the modification effect on the rheological properties of AP, and both the elastic property and viscosity value were promoted after thermal processing. The modification effect of viscoelastic properties (*G*′ and *G*″) was observed whne XG was incorporated into the composite formula. Increasing the XG ratio from 7:3 to 6:4 (AP:XG) triggers the rheological transformation from a liquid-like form to a solid-like state, and the viscosity value shows that the AP-XG composite system exhibits better thermal stability after thermal processing. The ambient modifiers of pH (pH < 4) and calcium chloride concentration (7.5%) with an optimal AP-XG ratio of 7:3 led to weak-gel-like behavior (*G*″ < *G*′), helping to maintain the texture properties of dysphagia-friendly features similar to those prior to the thermal processing.

## 1. Introduction

Aging can lead to dysphagia, a condition characterized by the degradation of swallowing function involving the oropharyngeal muscles and nerves [1,2,3]. The prevalence of dysphagia among the elderly in the Midwest of the United States ranges from 6% to 9%, but it is estimated to be 15–22% in community residents over 50 years old and as high as 40–60% among those in assisted living facilities and nursing homes [3]. Diseases such as stroke, Parkinson’s disease, multiple sclerosis, and neoplasms can also cause dysphagia [4,5]. To ensure the safe swallowing of flowable foods for individuals with dysphagia, thickeners are often used to modify the consistency of thin liquids, creating a safe-progressive alimentary bolus that can pass through the esophagus safely [6]. Consequently, thickeners serve as the foundation for creating dysphagia-friendly matrices for consistency adjustment [1,4,5,6].

A multi-disciplinary volunteer committee organized by the International Dysphagia Diet Standardisation Initiative (IDDSI) developed a general diet framework for with dysphagia [7]. The framework categorizes the consistency of food fluids into several levels by assessing a designated 10 mL syringe without a needle within a period of 10 s in contrast to a fork drip test [7,8]. The IDDSI framework is developed as per the guidelines issued by the most recognized organizations, such as the National Dysphagia Diet (NDD) proposed by the American Dietetic Association in 2002 [9], the Dysphagia Diet Committee of the Japanese Society of Dysphagia Rehabilitation [10,11] and the Australian Standardised Labels and Definitions [12]. These organizations also indicate that a fluidized food matrix bearing a viscosity greater than 50 mPa.s at a designated shear rate of 50 1/s is generally considered dysphagia-friendly [7,8,9,10,11,12].

The IDDSI framework, despite not including viscosity measurement, emphasizes the importance of food matrices having consistent apparent viscosity (thickness) at the specified test shear rate (50 1/s) to influence flow behavior and rheological characteristics [13,14,15,16]. While the IDDSI guideline is straightforward and accessible, it predominantly provides qualitative descriptions of food texture and liquid consistency. It lacks quantitative information on the textural and rheological characteristics of food and liquids. This gap makes it challenging for industry partners to ensure the quality of their modified diet products consistently [8]. The lack of quantitative data hinders researchers from exploring how food texture and liquid rheology affect swallowing [17]. Relying solely on qualitative assessments cannot capture the nuanced variations in food or liquids categorized within the same IDDSI level [18]. Consequently, there is a need to create quantitative metrics for the textural and rheological properties of food and liquids to enhance and support the IDDSI guidelines [17,18,19]. Cichero et al. highlighted the practical and scientific challenges in accessing rheological testing equipment and expertise needed for more comprehensive studies [7]. Furthermore, understanding the non-Newtonian and elastic behaviors of thickened food matrices requires careful observation and consideration using rheological methods beyond rotary viscometers [20,21,22,23].

Starch- and gum-based thickeners are commonly used to create dysphagia-friendly formulations that yield suitable fluidized matrices [24,25]. Starch-based thickened matrices may require additional carbohydrate calories to sustain primary energy metabolism [26]. However, these matrices are susceptible to viscosity reduction within a short 10 s window due to amylase-containing saliva during oral processing, posing a risk of accidental inhalation and aspiration pneumonia for dysphagic individuals [27,28,29]. The molecular degradation and consistency reduction of starch-based matrices due to the enzymatic activity of saliva alpha-amylase presents a significant concern. These changes can compromise the safety of the food bolus, increasing the risk of accidental inhalation and, consequently, aspiration pneumonia. Acknowledging these risks, it is clear that choosing non-starch-based (gum-based) matrices is a preferable approach. These alternatives are inherently more resistant to enzymatic breakdown, offering a more stable and predictable consistency. This stability is crucial in reducing the risk of aspiration and ensuring the safety and efficacy of dysphagia diets. Hydrocolloid gums are often added to stabilize the consistency of commercially available dysphagia-friendly products. Among these, xanthan gum is highly versatile, capable of forming networks, resistant to acidity, and can serve as an excipient when used alone or in combination with other polymers [30]. Yoon and Yoo studied the rheological behaviors of xanthan gum-based thickened formulas with 8 g of carbohydrate per 100 mL and found significant differences in shear-thinning behavior [31]. Ortega et al. reported that matrices with a consistency between 250 and 1000 mPa·s, achieved with a mixed starch and xanthan gum product, resulted in safe swallowing [32]. However, starch-based thickened matrices may face texture depletion during thermal processing, with significant viscosity reduction reported after simulated pasteurization [33]. Considering these challenges, gum-based matrices, such as pectin materials, have gained attention as thickeners, particularly when calorie supplementation is not a primary concern.

Pectin materials, found in plant cell walls, are chemically complex polysaccharides composed of various components, including homogalacturonans, rhamnogalacturonans-I, rhamnogalacturonans-II, xylogalacturonan, and apio-galacturonan moieties [34,35,36,37]. Their primary structure consists of repeating units of α-D-galacturonic acid linked with 1,4-glucosidic bonds and esterified with methyl residues at C-6 [36]. Commercial pectin products are classified as high- (>50%) or low- (<50%) methoxyl pectin based on their degree of methyl esterification (DE), with the European regulation requiring at least 65% α-D-galacturonic acid [36]. Pectin molecules can form three-dimensional aggregates, allowing them to create hydrocolloidal matrices with thickening properties [37]. These properties make them suitable as texture modifiers to meet dysphagia-friendly requirements [38]. Additionally, pectin materials offer health benefits beyond their rheological and mechanical properties, including antioxidant activities [34], anti-diabetic effects [39,40], and reductions in low-density lipoprotein [41]. They also serve as dietary fiber sources that function as prebiotics for intestinal fermentation [42,43,44]. We previously reported on the rheological and texture properties of a low-degree-of-esterification (DE = 45.4%) apple pectin-based composite formula with xanthan gum as a modifier, showing potential for dysphagia-friendly applications [33]. However, the effects of thermal processes on xanthan gum–apple pectin (XG–AP) composite formulas with varying DE levels of pectin materials remain unexplored and warrant investigation.

In this study, we used high-methoxyl apple pectin (AP) as the base thickener to develop thickened model matrices intended for commercial distribution within a sterilized logistics chain involving thermal processing that causes variations in rheological behavior. Therefore, the aims of this study are: (1) to investigate how blending xanthan gum (XG) and AP in varying proportions modifies rheological properties and determines the optimal thermal stability ratio; (2) to explore the modification effects and identify the optimal thermal stability formula by introducing calcium ions and adjusting pH in the XG–AP system; and (3) to assess the suitability of the XG–AP system as a thickening formula for dysphagia-friendly fluidized matrices.

## 2. Materials and Methods

### 2.1. Materials and Chemicals

The XG thickener used in this study was generously provided by a local food additive agency (Gemfont Co. Ltd., Taipei, Taiwan). XG is a long-chain polysaccharide (circa 2000 kDa) with d-glucose, d-mannose, and d-glucuronic acid as the main building skeleton in a nearly equal ratio in terms of quantitative molecule amounts with a high number of trisaccharide side chains associated with cations of sodium, potassium, and calcium [45,46]. The powdered apple pectin (AP) (Solgar, Inc., Leonia, NJ, USA) was purchased. Its degree of esterification was subjected to our in-house titration evaluation and reported in Section 3; we deemed it as presumably a high-methoxy AP. Calcium chloride anhydrous (CCA) as a divalent calcium source was obtained from Sigma-Aldrich Co. (St. Louis, MO, USA). A commercial powder, Neo-high Toromeal III^®^ (TRM, Food-care, Inc., Tokyo, Japan), was obtained as the control sample for the texture examination. Unless stated otherwise, all other chemicals in this study were of chemical grade, such as powdered citric acid and sodium bicarbonate.

### 2.2. Degree of Esterification (DE)

The DE of apple pectin (AP) was estimated as described previously [47] with minor modifications. Briefly, 0.2 g of powdered apple pectin was thoroughly dissolved in 200 mL of deionized water by stirring under ambient temperature for 2 h. Upon complete mixing, 3–5 drops of phenolphthalein indicator solution (C_20_H_14_O_4_, Sigma-Aldrich Co., St. Louis, MO, USA) were employed, and the dissolved AP solution was thereafter subjected to titration with 0.1 M NaOH until the mixture turned pink, and this lasted for 30 s; the depletion volume of NaOH (mL) was recorded as *V*_1_. Another 10 mL of NaOH was added and underwent stirring for 30 min. Subsequently, 10 mL of 0.1 M HCl was thoroughly mixed with the pink solution to neutralize the solution until it turned transparent, followed by a second titration of 0.1 M NaOH until the solution turned pink again for another 30 s. The second depletion volume of NaOH was recorded as *V*_2_. The DE was calculated according to Equation (1):(1)$$DE\%=\frac{V_{1}}{V_{1}+V_{2}}\times 100$$

### 2.3. Sample Preparation

To measure the basic characteristics of AP and to evaluate its optimal formula with possible modifications, sample matrices were prepared according to the formulae listed in Table 1. Sample powders were weighed and dispersed evenly in reverse osmosis water; to adjust the designated pH of the sample matrices, a small amount of reverse osmosis water was taken to dissolve the hydrocolloid completely, and then the pH was adjusted with a citric acid solution and sodium bicarbonate solution. The mixed matrix was stirred with an electronic stirrer at ambient temperature for complete dissolution, and the sample was poured into a 15 mL centrifuge tube and stored at 4 °C for future usage.

### 2.4. Thermal Processes

We employed a retort autoclave (SS-325, Tomy Seiko Co., Ltd., Tokyo, Japan) in simulating thermal processing. Control sample matrices (2 wt% AP) were sealed in designated centrifuge tubes equipped with temperature data loggers (MadgeTech, Inc., Warner, NH, USA). They were subjected to programmed treatments, including 5 or 10 min at 95 °C, 5 min at 105 °C, and 1 min at 115 °C. Appendix A illustrates the sample matrices’ actual temperature profiles over time, showing slight variations from the programmed temperature patterns due to thermal penetration delays. We selected 95 °C and 105 °C for 5 min as our thermal treatment settings, as these temperatures are commonly used for sterilizing high-acid foods. Pectin-based matrices typically have a pH below 4.6, categorizing them as high-acid (acidified) food matrices.

### 2.5. Rheological Measurements and Flow Behavior Characterization

A stress-controlled rheometer (DHR-2, TA Instruments, New Castle, DE, USA) equipped with a plate-and-plate stainless fixture (diameter = 40 mm, gap distance = 1000 μm) or a cylindrical double-gap fixture (inner cup diameter = 30.2 mm, inner motor diameter = 31.97 mm and outer diameter = 34.98 mm) was used. A Peltier temperature controller was used to maintain the temperature measurement at 25 °C. A solvent trap was employed to prevent moisture loss. The rheological measurements in chronological order are: (1) dynamic oscillatory strain sweeps to identify the linear viscoelastic region (LVR), (2) frequency sweeps with 2 min of rest duration after strain sweeps, (3) shear rate-dependent viscosities with 2 min rest duration after frequency sweep. The measurement parameters are listed in Table 2.

### 2.6. Texture Profile Analysis (TPA)

The authorized method of TPA issued in 2009 by the Japanese Ministry of Consumers, “Food for patients with swallowing difficulty”, under the regulation of “Food for special dietary uses”, was adopted for the characterization of model food matrices [35].

### 2.7. Data Manipulation and Statistical Analysis

The experimental data of this study were plotted and presented by Sigmaplot^®^ version 10.0. One-way ANOVA for LVR and loss tangent (tan *δ*) and two-way ANOVA for shear rate dependent viscosity were performed to determine significant differences using Duncan’s multiple range tests based on three replications; paired sample *t*-tests were utilized to justify the significance of differences among the mean values of textural properties based on nine replications. SPSS (Statistical Package for the Social Science) version 19.0 was used for the statistical analysis, and difference comparisons were made within the confidence interval of 95%, unless otherwise specified.

## 3. Results and Discussion

### 3.1. DE Evaluation of Apple Pectin Sample Powder

The AP sample used in this study exhibited a degree of esterification of 73.9 ± 1.00%, indicating its high-methoxyl form. However, achieving a weak-gel-like texture solely with high-methoxyl pectin is challenging due to its strict gel-forming conditions (requiring soluble solids >55% and pH < 3.5). Thus, the study explores composite formulas to create food matrices with the desired texture, with xanthan gum (XG) assistance. XG matrices generally exhibit elastic characteristics and contribute to the formation of thickened composite matrices (TCMs). Building on our previous study [35], we prepared two types of 2 wt% TCMs with different XG-to-AP ratios (4:6 and 3:7) to assess their rheological properties. Additionally, we prepared solitary (standalone) XG matrices using the same formula as a reference to evaluate the improvement in thermal stability brought about by TCMs.

### 3.2. Thermal Stability Analysis of Thickened Composite Matrices (TCM)

#### 3.2.1. Effects of Xanthan Gum (XG) and High-Methoxy Apple Pectin (AP) Composite Ratio on Rheological Behavior

Figure 1A(a) displays the strain sweep profiles (50 rad/s) of the composite XG–AP and standalone XG thickened matrices without thermal processing, both constrained to 2% weight. The 2% AP matrix exhibits liquid-like behavior (Appendix B), while standalone XG matrices (0.6% or 0.8%) exhibit solid-like behavior within the linear viscoelastic region (LVR), as summarized in Table 3. Increasing the XG ratio from XG0.6AP1.4 to XG0.8AP1.2 shifts the viscoelastic characteristics from liquid-like to solid-like. Figure 1A(b) presents frequency sweep profiles at 1% strain for the samples without thermal processing. The results indicate that the qualitative features, as exhibited by the slopes of G′ for standalone XG (XG0.6 and XG0.8) and composite XG–AP (XG0.6AP1.4 and XG0.8AP1.2) matrices, are similar, while the slopes of the G″ curve vary significantly. The overall trends suggest that the addition of AP to the composite XG–AP group imparts viscous characteristics to the samples, while XG primarily contributes to the elastic characteristics of the composite XG–AP matrices. Table 4 provides the viscosity values at different shear rates for all samples. The viscosity of each sample decreases as the shear rate increases, with the standalone XG group (XG0.6, XG0.8) exhibiting a larger degree of shear thinning compared to the composite groups XG0.6AP1.4 and XG0.8AP1.2.

To evaluate the modification effect of the XG composite ratio on the thermal stability of thickened matrices with the AP base, the set thermal processes for 5 min at 95 °C and 105 °C were imposed, and the results are presented in Figure 1B and Figure 1C, respectively. Comparing the results of solitary (standalone) XG samples (XG0.6 and XG0.8) in terms of strain sweep and frequency sweep, both *G*′ and *G*″ values decreased significantly after the thermal processing, and the increase in thermal processing temperature resulted in a more significant decrease. For composite matrices, the strain sweep profile of the 95 °C processing shows that the decreases in G′ and G″ in the XG0.6AP1.4 sample are less than those in the XG0.8AP1.2 sample. The viscoelastic characteristics of the former (XG0.6AP1.4) indicate good thermal stability at 95 °C for 5 min. However, the matrices treated at 105 °C (Figure 1C) show very different trends.

Compared with the results in Figure 1B(a) for the 95 °C processing, the sample XG0.8AP1.2 in Figure 1C(a) does not show a significant decrease in G′, but shows a substantial decrease in *G*″. Presumably, the *G*″ of the aforementioned composite matrices (0.8/1.2) is mainly dominated by apple pectin molecules, so the rise in temperature aggravates the degradation of apple pectin. It leads to the decline of composite matrices, which can also be observed from the results of the XG0.6AP1.4 sample. In the strain sweep profile, the XG0.6AP1.4 matrix shows liquid-like characteristics (*G*′ < *G*″) after thermal processing at 95 °C; however, the temperature rise would promote pectin degradation. As a result, the XG0.6AP1.4 sample shifted to a solid-like behavior (*G*″ < *G*′) shown in the strain sweep profile after the thermal processing at 105 °C. The frequency sweep results of thermal processing at 105 °C also indicate that the viscoelastic moduli of XG (control) group samples are lower than those of samples subjected to thermal processing at 95 °C due to the increase in temperature. In the composite group, because pectin is the main component contributing to the viscous characteristics, the slope of the *G*″ curve of XG0.6AP1.4 is significantly greater than that of XG0.8AP1.2, and both samples demonstrate more remarkable elastic characteristics with the increase in pectin degradation after the thermal processing.

Table 4 exhibits the shear-rate-dependent viscosity of the composite matrices and their corresponding controls prior to and after the designated thermal processing. All samples show a similar trend of shear thinning, while the standalone XG sample matrices show the most significant degree of shear thinning. For the XG0.8AP1.2 composite matrix, the degree of shear thinning is close to that of the standalone XG group due to its higher XG ratio.

Following a 5 min exposure to a 95 °C thermal load, the viscosity at a shear rate of 50 1/s decreased by varying amounts: 37.6% for XG0.6 (i.e., 280 → 174 mPa.s), 61.6% for XG0.6AP1.4, 8.5% for XG0.8, 39.2% for XG0.8AP1.2, and 54.1% for AP2.0. These results indicate that higher concentrations of standalone XG matrices correspond to more substantial viscosity reductions after thermal processing. Notably, XG itself does not provide resistance to thermal processing, aligning with findings reported by Naji et al. [48].

While the viscosity of AP matrices also decreased significantly post-thermal processing, composite matrices experienced only slight reductions. Notably, the XG0.6AP1.4 matrix exhibited the least viscosity reduction, at 8.5%. The composite XG–AP matrices present stronger thermal stability than the standalone XG or AP, and XG0.6AP1.4 (or 3:7 in ratio) is the best among them. This may be due to certain co-structures formed by XG and AP that impart the composite matrices’ excellent thermal stability. Therefore, the ideal modification effect can be achieved by exploring the optimized mixing ratio of XG and AP.

Deducing from the observations above, the remarkable rheological behavior provided by XG can be utilized to modify the viscoelastic characteristics of the AP matrices, which are dominated by viscous characteristics at low concentrations. In addition, the composite matrices can provide the corresponding hydrocolloidal matrices with a polymer system possessing more robust thermal stability. Therefore, including composite matrices into thickened edible fluid matrices for seniors is quite exploitable, and provides products featuring thermal stability after high-temperature processing.

#### 3.2.2. Modification Effects of pH and CCA on the Rheological Behavior of Thickened XG–AP Matrices


Preliminary tests


The pH of the 2% AP is approximately 3.3 (control), and Appendix B provides insights into the rheological behavior of the 2% AP matrices under varying pH and thermal conditions. Thahur et al. [49] have reported that carboxyl dissociation in AP matrices can be hindered under low-pH conditions. Following thermal processing at 95 °C and 105 °C, it is evident that the *G*″ for matrices at pH 5 and 6 are significantly lower compared to those at pH 3 and 4. This observation suggests that *β*-elimination reactions may occur most rapidly at pH levels near neutrality during the thermal pyrolysis of pectin, aligning with the findings of Fraeye et al. [50]. Additionally, high-methoxy pectin is generally considered resistant to acid hydrolysis. In summary, AP matrices subjected to 5 min thermal loading at 95 °C exhibit higher apparent viscosity compared to those at 105 °C. Furthermore, it can be inferred that maintaining a low pH can help mitigate thermal degradation in terms of consistency.

Appendix C illustrates the impact of calcium ions (Ca2.5 through 10) on standalone 2% AP matrices. The addition of CCA as the calcium ion source was expected to hydrate the water in the matrices, resulting in a denser intermolecular pectin structure. A prior study by Noriah et al. [51] noted that the addition of calcium ions to low-concentration pectin matrices led to an increase in *G*′, consistent with the findings in this study. While the loss tangent value for samples with added calcium ions is slightly higher, ~1 (Table 3), compared to the control samples, it can be inferred that incorporating CCA as the calcium ion source predominantly enhances the elastic properties of pectin m matrices rather than the viscosity properties, making the samples closer to a solid-like state. The experimental results, both before and after thermal processing, indicate that the formulations containing Ca2.5 and 7.5 exhibit more favorable modification effects, and are therefore selected for further formulation evaluation.
Effects of CCA Concentration

Previous experiments indicate that adding CCA as a calcium ion source can modify the viscoelastic characteristics of the 2% standalone AP matrices, and the samples added with calcium ions show higher viscosity (thickening effect) after thermal processing. Therefore, this section aims to explore the effect of calcium ion (CCA) dosage on the rheological properties of the samples. The groups with the highest viscosity after thermal processing (added with 2.5% and 7.5% CCA) in the previous experiment and the XG0.6AP1.4 matrix base without CCA addition were employed for the investigation. 

Figure 2A displays the rheological profiles of the three samples before thermal processing. The strain sweep (at a high frequency of 50 rad/s) reveals that the liquid-like state of XG0.6AP1.4 transitions to a solid-like state when calcium ions are added (XG0.6AP1.4-Ca2.5 and XG0.6AP1.4-Ca7.5), consistent with previous experimental findings that the addition of CCA enhances the elasticity of the matrices. The influence of CCA on XG–AP composite matrices is evident, maintaining the desirable weak-gel-like state pursued in this study. Notably, in the absence of CCA, the *G*″ of composite matrices exhibits significant changes after thermal processing (Figure 2A(a)). Furthermore, in frequency sweep tests, the samples with added CCA (Ca2.5 and Ca7.5) consistently exhibit significantly higher values of *G*′ and *G*″ compared to matrices in the absence of CCA.

It is hypothesized that calcium ions may enhance the molecular structure of composite matrices, making them more robust. When examining the frequency sweep profile, particularly in the high-frequency region, samples lacking CCA exhibit characteristics that are more viscous than elastic. Notably, a crossover point is observable in this context. However, the sample’s molecular structure is more stable after the addition of calcium ions, as the sample’s elasticity becomes dominant and there is no crossover point within the same range of frequencies. Therefore, the dominance of viscosity would occur only at still higher frequencies. In examining the impact of CCA addition, the results before thermal processing reveal that the XG0.6AP1.4-Ca2.5 and XG0.6AP1.4-Ca7.5 curves closely coincide in both strain and frequency sweep profiles. This suggests that calcium ions from CCA exert a limited modifying effect on the samples prior to heating. Therefore, a further investigation was conducted to assess whether calcium ion concentration would affect the rheological properties of the samples after thermal processing.

Figure 2B displays the rheological profiles of the samples subjected to a 5 min thermal treatment at 95 °C. In the strain sweep profile, it is observed that the *G*′ of XG0.6AP1.4-Ca2.5 is higher than that of XG0.6AP1.4-Ca7.5. However, the *G*″ of XG0.6AP1.4-Ca2.5 is slightly lower than that of XG0.6AP1.4-Ca7.5. This outcome aligns with previous experimental findings when comparing the impact of CCA addition on standalone 2% AP matrices, where Ca2.5 and Ca7.5 exhibited the most favorable modification effects prior to thermal processing.

However, after undergoing thermal processing at 95 °C for 5 min, it becomes evident that XG0.6AP1.4-Ca7.5 exhibits superior thermal stability compared to XG0.6AP1.4-Ca2.5. The latter, XG0.6AP1.4-Ca2.5, displays reduced thermal stability due to a decrease in G″, indicating a weakening of the pectin structure. This results in the XG–AP composite matrices leaning more towards a viscoelastic profile biased towards XG, and consequently experiencing an increase in elastic characteristics. A notable difference in the frequency sweep profile compared to the pre-thermal group is in the absence of the original crossover point in the high-frequency region for XG0.6AP1.4. This absence is attributed to the partial degradation of the pectin structure. The results obtained from the 5 min thermal processing at 95 °C highlight that a higher calcium ion concentration is effective.

Figure 2C shows the rheological profiles of the samples affected by a 5 min thermal load at 105 °C. Due to the increase in temperature compared to the temperature of 95 °C, XG0.6AP1.4-Ca7.5 presents almost the same rheological characteristics as XG0.6AP1.4-Ca2.5 in its strain sweep and frequency sweep results. As the temperature rises to105 °C, the increase in calcium ion concentration ceases to enhance the thermal stability. It has been reported that the addition of calcium ions can enhance the associative strength of high-methoxyl pectins [52,53], diverging from the “egg-box” model seen in low-methoxyl pectins and more closely resembling alginate gelation [54]. Yang et al. [55] have noted that a calcium-concentrating process, brought about by water evaporation, enables high-methoxyl pectin molecules to improve their hydrophobic interactions, which is crucial for gel formation. However, the specific molecular interactions between high-methoxyl pectin and calcium ions require further investigation to be fully understood. Therefore, the addition of CCA can improve the thermal stability of composite matrices only at a lower temperature (95 °C) under the same processing duration.

The viscosities (Table 4) measured at a shear rate of 50 Hz (1/s) were used to investigate the effect of CCA addition on the sample viscosity. The viscosity values of XG0.6AP1.4, XG0.6AP1.4-Ca2.5, and XG0.6AP1.4-Ca7.5 prior to thermal processing are 638.8, 757.6, and 807.7 mPa.s, respectively. They become lower after thermal processing at 95 °C, and even lower after thermal processing at 105 °C. The existence of calcium ions can promote the viscosity of the sample with or without thermal processing. According to the above results, the viscosity of XG0.6AP1.4-Ca7.5 is comparatively higher in all three conditions, although the disparity is small in general. We also observed that the inclusion of calcium ions into XG0.6AP1.4 composite matrices has a modification effect on both the viscoelastic characteristics and viscosities. Especially with Ca7.5, the viscoelastic characteristics after thermal processing at 95 °C are closer to the weak-gel-like state, and the viscosities are also the highest in all cases at a shear rate of 50 Hz (1/s). Therefore, XG0.6AP1.4-Ca7.5 is considered a better choice for the dysphagia-friendly fluid model system for use in the following evaluation.
Effects of pH

For the evaluation of the pH effect, an XG–AP composite ratio of 3:7 with 7.5% CCA was employed (i.e., XG0.6AP1.4-Ca7.5). The measured pH is 2.92 prior to any adjustment, which is in line with the low-pH environment. Therefore, samples with pH 5 and 6 are additionally prepared to investigate the influence of pH on the rheological properties and matrix stability after thermal processing.

Figure 3 presents the results of the rheological analysis concerning the pH effect. In Figure 3A, we observe the rheological profiles of the samples before thermal processing. The previous strain sweep results indicate that the incorporation of CCA substantially enhances the elasticity of XG0.6AP1.4 composite matrices, which originally exhibited liquid-like characteristics, to the extent that *G*′ surpasses *G*″ and demonstrates a weak-gel-like behavior. Notably, the samples with pH adjusted to 5 and 6 also exhibit a weak-gel-like pattern, but with both *G*′ and *G*″ values higher than those of the original XG0.6AP1.4-Ca7.5. 

The frequency sweep also shows results similar to the strain sweep, and additionally, the viscosities of the three samples exhibit shear-thinning characteristics. At a shear rate of 50 1/s, the measured viscosity follows the trend: control, XG0.6AP1.4-Ca7.5 (807.7 mPa.s) > pH 5, XG0.6AP1.4-Ca7.5 (565.0 mPa.s) > pH 6, XG0.6AP1.4-Ca7.5 (393.0 mPa.s).

Figure 3B shows the rheology profile of the same set of samples subjected to a 5 min thermal load at 95 °C. The strain sweep and frequency sweep profiles show that the *G*′ of the XG0.6AP1.4-Ca7.5 sample with unadjusted pH value (pH = 2.92, control) did not decrease significantly, and the sample still exhibited weak-gel-like characteristics. Thus, the composite matrices can maintain properties similar to those prior to thermal processing with good stability. In contrast, the other two samples with pH values adjusted to be close to the neutral one (pH = 7) showed quite different trends. The *G*′ and *G*″ pH 5, XG0.6AP1.4-Ca7.5 sample decreased, and especially the *G*″; the *G*″ of the pH 6, XG0.6AP1.4-Ca7.5 sample also decreased, while the *G*′ increased. 

Therefore, the above and previous experimental findings (as detailed in Appendix B) offer mutual support. When subjected to heating under pH 5 and 6 conditions, pectin undergoes more pronounced degradation compared to lower pH levels. This degradation similarly affects XG–AP composite matrices. The degradation of apple pectin, primarily responsible for imparting viscosity to the samples, results in a significant reduction in *G*″ for the pH 5 and 6 samples. It is speculated that the degree of AP degradation in the pH 6 environment exceeds that in the pH 5 environment, causing a weaker modification effect on XG–AP composite matrices in the latter. Consequently, both samples exhibit increased *G*′, resembling the viscoelastic traits of XG. 

In Figure 3C, which showcases the rheological profiles of the samples after thermal processing at 105 °C for 5 min, the overall trend remains largely consistent with that observed after thermal processing at 95 °C. As a higher temperature will accelerate pectin degradation, a decrease in G″ can be seen from the strain sweep profile of the XG0.6AP1.4-Ca7.5 sample (Figure 3C(a)). For comparison, the pH 5, XG0.6AP1.4-Ca7.5 sample exhibited a similar increase in the elastic characteristics as the pH 6, XG0.6AP1.4-Ca7.5 sample when the thermal processing temperature was increased to 105 °C.

The changes in viscosity due to thermal processing are calculated from the reported data shown in Table 4. For the measured viscosity at a shear rate of 50 1/s, the viscosity of all samples decreased after thermal processing. However, it can be seen that the two samples with pH close to the neutral state exhibited more pronounced viscosity reductions than those of lower pH values. After being heated at 95 °C and 105 °C for 5 min, the viscosity of the control, XG0.6AP1.4-Ca7.5 sample decreased by 16.24% and 24.9% (807.7 → 676.5 and 807.7 → 606.3 mPa.s), respectively; the pH 5, XG0.6AP1.4-Ca7.5 sample decreased by 57.8% and 62.1% (565.0 → 238.3 and 565.0 → 214.1 mPa.s), respectively; the pH 6, XG0.6AP1.4-0Ca7.5 sample decreased by 45.8% and 52.4% (393.0 → 213.1 and 393.0 → 186.9 mPa.s), respectively.

As per the previous discussions on the effects of the pH values of the matrix systems, the viscoelasticity and viscosity of the XG–AP composite matrices can be observed to decrease more significantly after thermal processing, with pH closer to the neutral state (pH 5 or 6). The degradation of apple pectin in these samples leads to decreased viscous characteristics, resulting in promoted elastic characteristics (similar to the rheological characteristics of XG). The viscosity, which currently serves as the primary standard [9,10,13,56] of dysphagia-friendly edible fluids, decreased significantly and most notably. Therefore, it can be deduced that the pH value is crucial when preparing edible fluid models. When high-methoxy pectin is used as the major thickener base, better thermal stability can be ensured with acid-base ones.

### 3.3. Summary of Rheological Behavior of Sample Matrices

Cichero [56] and Burning [57] highlighted a shift from starch-based dysphagia products to gum-based ones, citing improved palatability, stable thickness, and patient preference for serving temperature. Leonard et al. conducted a clinical trial with 118 enrolled dysphagia patients, comparing gum-thickened matrices to thin water and starch-thickened ones, all designated as having the consistency of nectar [58]. They found that gum-based administration resulted in reduced aspiration difficulty and lower PAS ratings [55]. Additional studies explored gum-based thickeners, focusing on amylase-resistance properties [27,28]. A study on 120 post-stroke patients with dysphagia showed that gum-based thickeners with a viscosity range of 150 to 800 mPa.s significantly improved swallowing safety [59]. Burnip [57] and Cichero [56] noted that applying these recommendations clinically may be limited due to the viscosity labeling system’s requirement of laboratory-grade rheometers and engineering expertise.

Chemical β-elimination, a major nonenzymatic degradation in pectin, involves hydrogen and glycosidic residue removal due to heating [60]. Other known pectin degradation methods include acid hydrolysis and demethylation [60,61]. Studies indicate that alkalized pectin allows β-elimination [62] and demethylation [63]. Diaz et al. found that β-elimination increases with higher temperatures, with an activation energy of 136 kJ/mol at pH 4.5, suggesting temperature dependency [60]. Higher temperatures promote β-elimination, while increased pH promotes demethylation [64,65]. Our results in the previous section align with these findings, especially at higher pH values (5 and 6) and with thermal treatment. 

Calcium chloride, designated as E-509 [66], is an approved food additive in the European Union, authorized for its sequestrant and firming agent properties. When fortifying food matrices with various calcium sources, moisture absorption during storage is a concern. To ensure experimental precision, we opted for calcium chloride anhydrous (CCA) as the calcium ion source due to its resistance to moisture absorption during storage. According to the manufacturer’s specifications, CCA exhibits a solubility of up to 74.5 g/100 mL in water at ambient temperature. We utilized a 40 g/100 mL solution for shear-rate-dependent viscosity measurements as another control. Notably, the calcium ion concentrations range from Ca2.5 to Ca40 and span from 0.91% to 14.55% (wt).

Table 3 gathers the results of G′, G″, and tan δ of different TCMs; a high versatility of these systems can be observed. The as-prepared matrices vary in behavior, and range from an almost creamy matrix (G′ or G″ lower than 20 Pa; tan δ > 0.7) to a weak-gel-like system (tan δ < 0.25), comparable to a previously reported multiple-layered w/o/w emulsion system [67]. Liu et al. [68] investigated the rheological and structural properties of different acidified (pH 4.8–6.2) and rennet-treated milk gels and reported that no data were obtained from their milk gel samples, which gave high tan δ values >1.0 and low G′ < 1. They highlight that the tan δ values of milk gel samples with renneting extents of 55% and 74% at pH 6.2 were identical (0.27); as the pH progressively reduced and became as low as 4.8 due to acidification, the tan δ values were promoted to 0.72 and 0.92, respectively. 

Based on the frequency sweep spectra, Ross-Murphy [69] demonstrated that the viscoelasticity of flowable matrices is generally classified into dilute and concentrated solutions and real gels. We noted that the standalone 2% apple pectin-based (AP2.0) matrices present a dilute solution pattern regardless of the thermal treatments, pH, and CCA fortification (Figure A2B(b) and Figure A3C(b)). In contrast, the composite XG–AP matrices present gel behavior (Figure 1(b), Figure 2(b) and Figure 3(b)) regardless of the thermal conditions, pH, and CCA fortification, indicating the modification effect of XG. 

XG-based thickeners like ThickenUP Clear^®^ (Nestlé Health Science, Vevey, Switzerland) are commonly used for individuals with dysphagia. Rofes et al. [28] found that adding 2.4 g of this commercial XG-based thickening formula to 100 mL of mineral water can create a honey-like viscosity. Our XG matrices, at 0.8 g/100 mL before thermal treatment, have a honey-like viscosity (351–1750 mPa.s). This similarity highlights the need for the careful consideration of source variations in dysphagia oral administration. Additionally, the frequency-dependent rheological moduli of XG0.6AP1.4-Ca7.5 following thermal treatments under controlled pH (Figure 3(b)) converge at 100 1/s due to a G′ dip. This behavior may result from β-elimination in the pectin and the negatively charged nature of XG, while 1 mg/mL XG, with or without 1 mg/mL Na-caseinate, was prepared within the pH 2.7–6.6 range, after incubation at 37 °C for 60 min [70]. In addition to interpreting the rheological behavior, we conducted a comprehensive statistical analysis of shear-rate-dependent viscosities, as presented in Table 4. Generally, a more rigorous thermal treatment (indicated by lowercase letters) resulted in higher viscosity levels (*p* < 0.05), except for the case of XG0.6AP1.4 at a shear rate of 10 1/s. It is worth emphasizing that at the composite ratio of 3/7, these matrices exhibited the best thermal stability, or, equivalently, the least variation in viscosity. The influence of pH values on the AP2.0 matrices is indicated by uppercase letters. Specifically, pH 5 and 6 led to significantly lower viscosity levels (*p* < 0.05) compared to the control at pH 3. In contrast, at pH 4, intensive thermal treatment also resulted in a significant viscosity reduction, rendering it unsuitable for claiming dysphagia-friendly status. With the incorporation of CCA into AP2.0, viscosity also experienced a significant increase (*p* < 0.05), although its overall impact remained statistically insignificant (*p* > 0.05). When varying the composite ratio of thickeners, changes in viscosity were found to be shear-rate-dependent, with no clear trends across the entire range (indicated as numbers). However, considering that the typical shear rate associated with human swallowing is approximately 50 1/s, it is noteworthy that viscosity within the range of 30 to 70 1/s consistently followed the trend of XG0.8AP1.2 = XG0.6AP1.4 > XG0.8 > XG0.6 (*p* < 0.05). Hence, it is deducible that pectin not only efficiently modulates the viscosity of XG-based hydrocolloids, but also acts as a beneficial dietary fiber source.

Our previous study [33] reported on the rheological behavior of distilled water and acidic orange juice, each individually thickened with tapioca starch and xanthan gum (XG) to achieve a nectar-like consistency. These beverages exhibited distinct rheological responses following a 6 min thermal treatment at 80 °C. Specifically, the tapioca-thickened distilled water consistently exhibited a viscous behavior after the thermal treatment, while the tapioca-thickened orange juice initially displayed an elastic behavior before transitioning to a viscous behavior after treatment. Furthermore, we observed that the XG-thickened samples consistently exhibited an elastic behavior in an aqueous base (distilled water or orange juice) before or after the thermal treatment. Our current study subjected the XG–AP composite matrices to more severe thermal loads of 95 °C and 105 °C. This further confirmed that XG allows the XG–AP composite matrices to consistently exhibit elastic dominance features, regardless of the pH condition and calcium content, even though the standalone AP matrices at 2% concentration exhibited viscous dominance features.

### 3.4. Texture Analysis

Under various thermal processing conditions, prior rheological measurements showed that the thickened XG–AP model matrix has good thermal stability under the processing condition of 95 °C for 5 min. The texture analysis results of samples prior to and after heating under the processing condition of 95 °C for 5 min were further compared, as shown in Table 5. The hardness of the XG0.6AP1.4-Ca7.5 sample prior to thermal processing was higher than that of other samples. However, after thermal processing, the hardness decreased significantly (*p* < 0.01). It can be deduced that the strength of the network structure of the samples significantly decreased after thermal processing. The adhesiveness of XG0.6AP1.4-Ca7.5 prior to and after the thermal processing shows only a slight decrease. Overall, it seems that thermal processing had little effect on the adhesiveness of the samples investigated.

The cohesiveness of all samples decreased after thermal processing, and there were significant differences. It can be inferred that after thermal processing, the bonding strength of the internal structure of the samples will decrease. According to the Japanese Society of Dysphagia Rehabilitation, foods are designated from 0.2 to 0.9; the measured results of the samples (prior to or after thermal processing) in this study are all within the standard range. The gumminess values prior to and after thermal processing for the XG0.6AP1.4-Ca7.5 sample show a very significant difference (*p* < 0.01). However, both gumminess values are greater than those with the same XG–AP ratio as the thickeners and a low-methoxy AP (DE = 45.2%) [35]. However, in our previous study, we measured a commercial powder product, Neo-high Toromeal III^®^ (TRM, Food-care, Inc., Tokyo, Japan), with the recommended concentration of 1.5% [35], as the control sample for texture profile analysis. The gumminess observed in this case (282 N/m^2^) was similar to that found in the current study. The measured gumminess after thermal processing was reasonable because the Japanese Society of Dysphagia Rehabilitation Foods standard [10] recommends that hardness to fall in the range of 300–10,000 N/m^2^, and that the maximum adhesiveness be less than 1500 J/m^3^. The current results show that the composite formula of XG0.6AP1.4-Ca7.5 meets these standards. Therefore, the proposed approach of adding pectin as the thickener seems reasonable for the matrix required for further thermal processing.

In this study, we found that the AP sample had a DE value of 73.9%, while in previous reports, the DE value of the AP sample was noted as 45.2%. In both instances, XG effectively functioned as a texture modifier. To mitigate the issues arising from material variations or complexity, adherence to existing standardization measures is recommended. For instance, the texture analysis standards provided by the Japanese Society of Dysphagia Rehabilitation Foods [10] are a reliable indicator of product specification and acceptance. To manage this complexity and ensure a robust and standardized formulation, we have conducted preliminary studies focusing on crucial formulation parameters, notably the calcium concentration and the ambient pH. These studies are essential for optimizing the pectin’s viscoelastic characteristics. Additionally, we have investigated the formulation ratio of xanthan gum, a known consistency modifier, to complement the pectin matrix. This approach allows us to fine-tune the texture and viscosity of the preparation, thereby addressing potentially acceptable consistency. By systematically studying these parameters and their interactions, we have developed a formulation: the ambient modifiers of pH (pH < 4) and calcium chloride concentration (7.5%) with an optimal AP–XG ratio of 7:3 led to weak-gel-like behavior (G″ < G′) that helps to maintain the texture properties of dysphagia-friendly features similar to those prior to the thermal processing. The IDDSI framework does not include a direct measurement of viscosity because it is an empirical method. The framework is designed to be a guideline for nursing professionals preparing food bolus with a dysphagia-friendly consistency to serve individuals with dysphagia over the duration of a certain meal. However, formulae and standards such as the Japanese Society of Dysphagia Rehabilitation Foods are available [10]. The flowable matrices still need to be defined and classified by specified shear-rate-dependent (10–100 1/s) viscosities [13], as we listed in Table 4. Our motivation for assessing rheology is related to the pursuit of scale-up potential, including extended shelf life with possible thermal processing for microbial/sanitation control and possible unit operation dissolving, mixing, or aseptic heat exchanging in terms of the viscoelastic characteristics. Chemical β-elimination, and other degradation processes such as acid hydrolysis and demethylation, affect pectin, especially under conditions of higher temperature and pH. Our studies align with previous findings, showing that these conditions promote pectin’s degradation. However, our composite XG–AP matrices, particularly XG0.6AP1.4, demonstrated superior thermal stability with minimal viscosity reduction, likely due to the presence of stabilizing co-structures between XG and AP. Optimizing the XG and AP composite ratio is key to enhancing the thermal stability of pectin-based matrices. We have previously reported on the co-structure of XG–AP composite matrices [35]. The composite thickener (AP1.8XG0.2) demonstrated smoother surfaces compared to the calcium ion sample (AP2Ca5) and other single thickener formulas through scanning electron microscopic image observations; additional research is needed to gain deeper insights. Starch-based matrices, often used in formulations for individuals with dysphagia, tend to decrease in viscosity when exposed to amylase in saliva. This reduction can increase the risk of accidental inhalation and aspiration pneumonia. Addressing this concern aligns with our research goals. We experimented with composite XG–AP matrices, selected for their unique structures. These matrices lack the alpha-1,4 glucosidic bonds usually present in starch, which are key to ensuring they do not pose inhalation or aspiration risks for those with dysphagia. Generally, significant viscosity depletion highlights the limitations of starch-based matrices. Therefore, our non-starch-based composite XG–AP formula could offer a more reliable alternative, given its better stability and alpha-amylase resistance for developing dysphagia-friendly matrices. 

## 4. Conclusions

This study demonstrated that the inclusion of XG can effectively modify the original AP matrices’ rheological properties, and substantially extend the accessible range of viscoelastic properties, resulting in ubiquitous weak-gel-like characteristics, as with commercially available geriatric foods. Furthermore, regarding thermal stability, the change in viscosity of the composite matrices after thermal processing is smaller than that of the standalone AP or XG matrices, and the group with AP:XG = 7:3 showed the best performance. The formula with 7.5% CCA showed the optimal viscoelastic characteristics after thermal processing. Especially with the addition of calcium ions, the viscoelastic characteristics of the XG0.6AP1.4 matrices were shown to change from liquid-like to solid-like for the full range of frequencies investigated—an ideal rheological feature for the further fortification of the minor nutrients required for individuals with dysphagia. 

## Figures and Tables

**Figure 1 foods-13-00090-f001:**
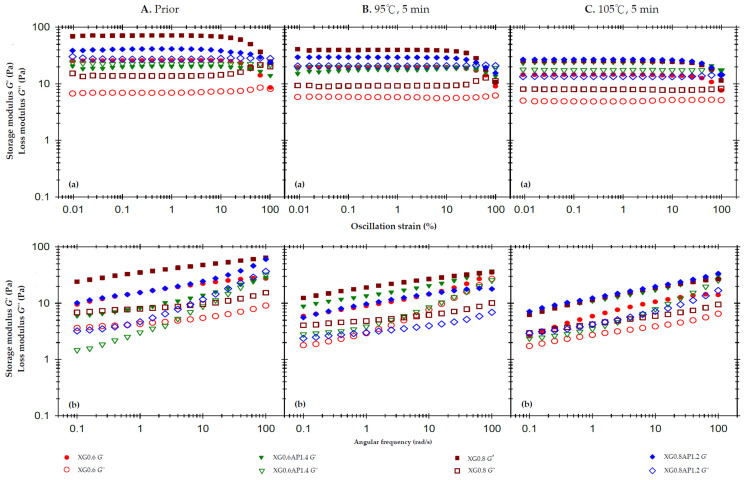
Storage modulus (*G*′) and loss modulus (*G*″) of composite xanthan gum and high-methoxy apple pectin (XG-AP) formula and the corresponding standalone XG thickened matrices based on a 2% weight constraint experiencing different thermal processing conditions: (**A**). prior to thermal processes, (**B**). after a 95 °C for 5 min process and (**C**). after a 105 °C for 5 min. (**a**) oscillatory strain sweeps at the frequency of 50 rad/s, and (**b**) angular frequency sweeps at 1% strain.

**Figure 2 foods-13-00090-f002:**
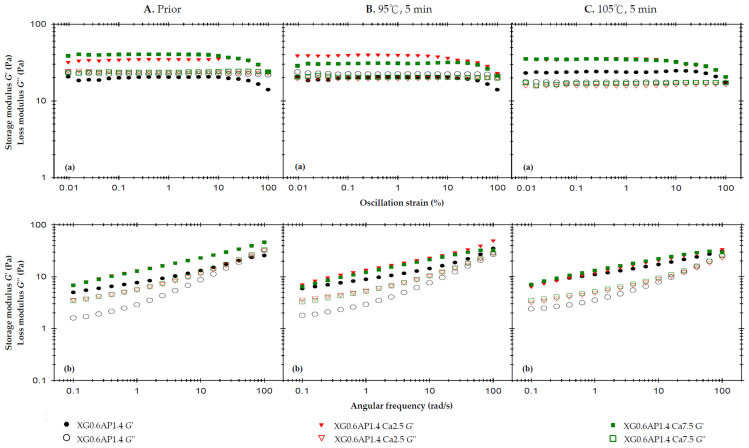
The effects of 2.5% and 7.5% calcium chloride anhydrous on storage modulus (*G*′) and loss modulus (*G*″) of the composite XG–AP (XG0.6AP1.4) thickened matrix experiencing different thermal processing conditions: (**A**) prior to thermal process, (**B**) after a 95 °C for 5 min process and (**C**) after a 105 °C for 5 min process with (**a**) oscillatory strain sweeps at the frequency of 50 rad/s, and (**b**) angular frequency sweeps at 1% strain.

**Figure 3 foods-13-00090-f003:**
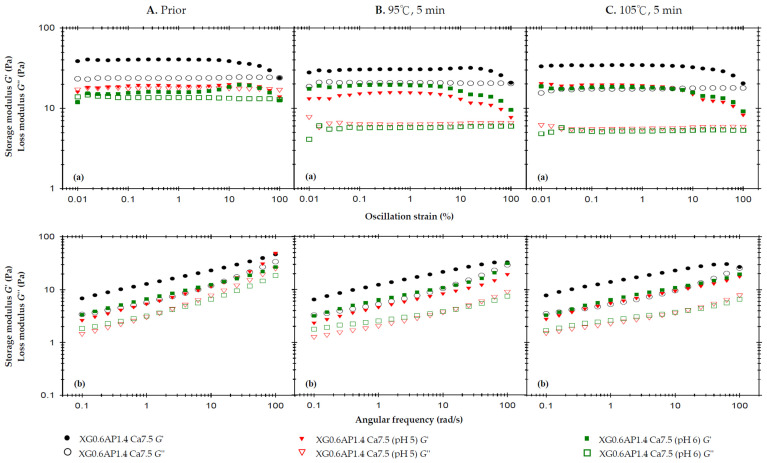
The effects of the pH of 2.5% and 7.5% calcium chloride anhydrous on storage modulus (G′) and loss modulus (G″) of the composite XG0.6AP1.4 thickened matrix with 7.5% calcium chloride anhydrous addition (XG0.6AP1.4-Ca.7.5) experiencing different thermal processing conditions: (**A**) prior to thermal process, (**B**) after a 95 °C for 5 min process and (**C**) after a 105 °C for 5 min process with (**a**) oscillatory strain sweeps at the frequency of 50 rad/s, and (**b**) angular frequency sweeps at 1% strain.

**Table 1 foods-13-00090-t001:** List of formulating parameters and groups of hydrocolloidal matrices with their corresponding codes.

Group	Matrix Code	Concentrations/pH or Ca^+2^
Preliminary		
Control	--	AP = 2 wt%
pH adjusted	pH3, pH4, pH 5, and pH6	AP = 2 wt%/Acid or alkaline
Ca^+2^ addition	Ca2.5, Ca5, Ca7.5, and Ca10	AP = 2 wt%/calcium chloride anhydrous (CCA)
Composite		
Composite ratio	XG0.6AP1.4 XG0.8AP1.2	XG = 0.6 wt% and AP = 1.4 wt% XG = 0.8 wt% and AP = 1.2wt%
Solitary XG	XG0.6 XG0.8	XG = 0.6 wt% XG = 0.8 wt%
Modifier		
Ca^+2^	XG0.6AP1.4-Ca2.5 XG0.6AP1.4-Ca7.5	XG = 0.6 wt% and AP = 1.4 wt% with 2.5 wt% CCA XG = 0.6 wt% and AP = 1.4 wt% with 7.5 wt% CCA
pH	[pH5, XG0.6AP1.4-Ca2.5] [pH6, XG0.6AP1.4-Ca2.5] [pH5, XG0.6AP1.4-Ca7.5] [pH6, XG0.6AP1.4-Ca7.5]	Corresponding CCA level with pH adjustment

**Table 2 foods-13-00090-t002:** Experimental parameters for the employed rheometer.

Experiment	Temperature (°C)	Strain (%)	Frequency (rad/s)	Shear Rate (1/s)
Strain sweep	25	0.01-100	50	-
Frequency sweep	25	1	0.1-100	-
Shear-rate-dependent viscosity	25	-	-	0.1, 0.3, 1, 3, 10, 30, 50, 70, and 100

**Table 3 foods-13-00090-t003:** Linear viscoelastic region data resulted from the strain sweep due to different thermal processes. Storage modulus (*G*′_LVR_); loss modulus (*G*″_LVR_); loss tangent (tan *δ*_LVR_) ^†^.

Treatment	Sample	*G*′_LVR_ (Pa)	*G*″_LVR_ (Pa)	tan *δ*_LVR_
Prior	XG0.6	25.63 ± 0.12 ^c^	6.94 ± 0.14 ^d^	0.27 ± 0.01 ^c^
	XG0.6AP1.4	19.91 ± 0.70 ^d^	22.55 ± 0.35 ^b^	1.13 ± 0.05 ^a^
	XG0.8	71.06 ± 0.31 ^a^	13.77 ± 0.11 ^c^	0.19 ± 0.00 ^d^
	XG0.8AP1.2	40.69 ± 0.64 ^b^	27.33 ± 0.16 ^a^	0.67 ± 0.0145 ^b^
95 °C	XG0.6	19.55 ± 0.08 ^c^	5.78 ± 0.11 ^d^	0.30 ± 0.01 ^c^
	XG0.6AP1.4	17.67 ± 0.76 ^d^	19.55 ± 0.16 ^b^	1.11 ± 0.06 ^a^
	XG0.8	39.63 ± 0.05 ^a^	9.20 ± 0.02 ^c^	0.23 ± 0.00 ^d^
	XG0.8AP1.2	29.36 ± 0.12 ^b^	20.55 ± 0.06 ^a^	0.70 ± 0.00 ^b^
105 °C	XG0.6	14.49 ± 0.06 ^d^	4.88 ± 0.04 ^d^	0.34 ± 0.00 ^c^
	XG0.6AP1.4	23.97 ± 0.38 ^c^	17.53 ± 0.10 ^a^	0.73 ± 0.01 ^a^
	XG0.8	25.25 ± 0.06 ^b^	7.97 ± 0.05 ^c^	0.32 ± 0.00 ^d^
	XG0.8AP1.2	26.84 ± 0.05 ^a^	13.42 ± 0.06 ^b^	0.50 ± 0.00 ^b^
Prior	XG0.6AP1.4	19.91 ± 0.70 ^c^	22.55 ± 0.35 ^c^	1.13 ± 0.05 ^a^
	XG0.6AP1.4Ca2.5	34.49 ± 0.46 ^b^	24.26 ± 0.13 ^a^	0.70 ± 0.01 ^b^
	XG0.6AP1.4Ca7.5	40.25 ± 0.30 ^a^	23.74 ± 0.22 ^b^	0.59 ± 0.01 ^c^
95 °C	XG0.6AP1.4	17.67 ± 0.76 ^c^	19.55 ± 0.16 ^c^	1.11 ± 0.06 ^a^
	XG0.6AP1.4Ca2.5	39.35 ± 0.41 ^a^	19.73 ± 0.17 ^b^	0.50 ± 0.01 ^c^
	XG0.6AP1.4Ca7.5	30.21 ± 0.49 ^b^	20.75 ± 0.17 ^a^	0.69 ± 0.02 ^b^
105 °C	XG0.6AP1.4	23.97 ± 0.38 ^c^	17.53 ± 0.10 ^a^	0.73 ± 0.01 ^a^
	XG0.6AP1.4Ca2.5	35.47 ± 0.53 ^a^	16.00 ± 0.10 ^c^	0.45 ± 0.01 ^c^
	XG0.6AP1.4Ca7.5	34.24 ± 0.19 ^b^	17.37 ± 0.12 ^b^	0.51 ± 0.00 ^b^
Prior	XG0.6AP1.4Ca7.5	40.25 ± 0.298 ^a^	23.74 ± 0.22 ^a^	0.59 ± 0.01 ^c^
	XG0.6AP1.4Ca7.5 (pH 5)	19.04 ± 0.42 ^b^	17.92 ± 0.15 ^b^	0.94 ± 0.03 ^a^
	XG0.6AP1.4Ca7.5 (pH 6)	15.72 ± 0.46 ^c^	13.79 ± 0.33 ^c^	0.88 ± 0.04 ^b^
95 °C	XG0.6AP1.4Ca7.5	30.21 ± 0.49 ^a^	20.75 ± 0.17 ^a^	0.69 ± 0.02 ^a^
	XG0.6AP1.4Ca7.5 (pH 5)	15.31 ± 0.45 ^c^	6.37 ± 0.10 ^b^	0.42 ± 0.02 ^b^
	XG0.6AP1.4Ca7.5 (pH 6)	19.38 ± 0.23 ^b^	5.74 ± 0.03 ^c^	0.30 ± 0.00 ^c^
105 °C	XG0.6AP1.4Ca7.5	34.24 ± 0.19 ^a^	17.37 ± 0.12 ^a^	0.51 ± 0.00 ^a^
	XG0.6AP1.4Ca7.5 (pH 5)	19.54 ± 0.22 ^b^	5.56 ± 0.03 ^b^	0.28 ± 0.00 ^b^
	XG0.6AP1.4Ca7.5 (pH 6)	18.07 ± 0.33 ^c^	5.26 ± 0.16 ^c^	0.29 ± 0.01 ^b^

^†^ Means ± SD of triplicate analyses are given. ^a–d^ Means with different lowercase superscripts within an individual thermal treatment (sub-column) differ significantly.

**Table 4 foods-13-00090-t004:** Shear-rate-dependent (10–100 1/s) viscosity (mPa·s) due to the thermal processing for different composite formulations with dysphagia-friendly potential ^†^.

Formula ^†^	Processing	Modifier	Shear Rate (1/s)				
			10	30	50	70	100
XG0.6	Prior	--	1036.8 ± 0.76 ^3, a^	411.9 ± 0.63 ^3, a^	280.4 ± 0.67 ^3, a^	222.0 ± 1.41 ^3, a^	178.7 ± 0.78 ^4, a^
	95 °C, 5 min	--	755.8 ± 0.60 ^b^	272.1 ± 0.14 ^b^	174.8 ± 0.18 ^b^	129.9 ± 0.09 ^b^	95.1 ± 0.09 ^b^
	105 °C, 5 min	--	615.3 ± 0.23 ^c^	228.6 ± 0.05 ^c^	142.0 ± 0.03 ^c^	103.9 ± 0.01 ^c^	74.8 ± 0.01 ^c^
XG0.6AP1.4	Prior	--	1327.7 ± 3.91 ^2, b^	808.3 ± 1.55 ^1, a^	638.7 ± 0.51 ^1, a^	549.4 ± 0.59 ^1, a^	465.6 ± 0.43 ^1, a^
	95 °C, 5 min	--	1299.2 ± 4.09 ^c^	754.7 ± 1.50 ^b^	584.6 ± 0.61 ^b^	496.5 ± 0.81 ^b^	416.2 ± 0.71 ^b^
	105 °C, 5 min		1376.0 ± 4.89 ^a^	753.7 ± 1.58 ^c^	568.9 ± 0.73 ^c^	476.7 ± 0.72 ^c^	394.0 ± 0.74 ^c^
XG0.8	Prior	--	2542.4 ± 3.84 ^1, a^	1060.1 ± 2.00 ^2, a^	732.9 ± 0.85 ^2, a^	576.6 ± 1.21 ^2, a^	450.2 ± 0.94 ^3, a^
	95 °C, 5 min	--	1145.7 ± 1.96 ^b^	430.2 ± 0.40 ^b^	281.2 ± 0.55 ^b^	217.2 ± 0.46 ^b^	162.2 ± 0.48 ^b^
	105 °C, 5 min	--	860.1 ± 0.22 ^c^	319.6 ± 0.10 ^c^	200.7 ± 0.04 ^c^	147.1 ± 0.04 ^c^	105.6 ± 0.02 ^c^
XG0.8AP1.2	Prior	--	2180.0 ± 4.92 ^1, a^	1167.3 ± 1.88 ^1, a^	870.1 ± 0.80 ^1, a^	719.1 ± 0.83 ^1, a^	583.8 ± 0.71 ^2, a^
	95 °C, 5 min	--	1376.9 ± 3.87 ^b^	718.4 ± 1.38 ^b^	528.8 ± 0.70 ^b^	433.8 ± 0.67 ^b^	349.6 ± 0.64 ^b^
	105 °C, 5 min	--	1204.3 ± 2.48 ^c^	555.4 ± 0.95 ^c^	385.1 ± 0.48 ^c^	305.6 ± 0.46 ^c^	239.4 ± 0.43 ^c^
AP2.0	Prior	control	290.8 ± 0.27 ^β, A, a^	270.3 ± 0.14 ^β, A, a^	258.1 ± 0.12 ^β, A, a^	249.2 ± 0.10 ^β, A, a^	238.5 ± 0.06 ^β, A, a^
	95 °C, 5 min	control	123.4 ± 0.21 ^b^	120.4 ± 0.13 ^b^	118.5 ± 0.12 ^b^	116.5 ± 0.08 ^b^	113.8 ± 0.09 ^b^
	105 °C, 5 min	control	55.0 ± 0.14 ^c^	53.6 ± 0.11 ^c^	53.3 ± 0.09 ^c^	53.0 ± 0.06 ^c^	52.6 ± 0.04 ^c^
	Prior	pH = 3	270.6 ± 1.04 ^A, a^	253.4 ± 0.33 ^A, a^	243.2 ± 0.21 ^A, a^	235.3 ± 0.14 ^A, a^	225.5 ± 0.08 ^A, a^
	95 °C, 5 min	pH = 3	116.1 ± 0.22 ^b^	112.6 ± 0.20 ^b^	110.6 ± 0.18 ^b^	108.8 ± 0.16 ^b^	106.4 ± 0.00 ^b^
	105 °C, 5 min	pH = 3	48.8 ± 0.11 ^c^	47.1 ± 0.03 ^c^	46.6 ± 0.03 ^c^	46.3 ± 0.02 ^c^	46.0 ± 0.01 ^c^
	Prior	pH = 4	247.9 ± 0.70 ^A, a^	233.6 ± 0.25 ^A, a^	223.7 ± 0.11 ^A, a^	216.0 ± 0.12 ^A, a^	206.7 ± 0.07 ^A, a^
	95 °C, 5 min	pH = 4	119.2 ± 0.17 ^b^	115.6 ± 0.10 ^b^	113.1 ± 0.08 ^b^	111.0 ± 0.04 ^b^	108.1 ± 0.05 ^b^
	105 °C, 5 min	pH = 4	61.9 ± 0.18 ^c^	60.9 ± 0.09 ^c^	60.4 ± 0.09 ^c^	60.0 ± 0.07 ^c^	59.4 ± 0.05 ^c^
	Prior	pH = 5	257.2 ± 0.53 ^B, a^	237.5 ± 0.42 ^B, a^	224.8 ± 0.38 ^B, a^	214.8 ± 0.12 ^B, a^	203.0 ± 0.28 ^B, a^
	95 °C, 5 min	pH = 5	60.8 ± 0.18 ^b^	60.1 ± 0.09 ^b^	59.8 ± 0.08 ^b^	59.5 ± 0.07 ^b^	59.0 ± 0.02 ^b^
	105 °C, 5 min	pH = 5	17.4 ± 0.13 ^c^	17.3 ± 0.05 ^c^	17.2 ± 0.04 ^c^	17.2 ± 0.01 ^c^	17.2 ± 0.02 ^c^
	Prior	pH = 6	240.5 ± 0.16 ^B, a^	225.6 ± 0.08 ^B, a^	215.5 ± 0.07 ^B, a^	207.2 ± 0.05 ^B, a^	197.0 ± 0.03 ^B, a^
	95 °C, 5 min	pH = 6	11.4 ± 0.10 ^b^	11.1 ± 0.03 ^b^	11.0 ± 0.02 ^b^	10.9 ± 0.00 ^b^	10.9 ± 0.02 ^b^
	105 °C, 5 min	pH = 6	5.91 ± 0.05 ^c^	5.49 ± 0.01 ^c^	5.39 ± 0.00 ^c^	5.3 ± 0.00 ^c^	5.2 ± 0.00 ^c^
AP2.0	Prior	Ca2.5	721.9 ± 0.64 ^α, a^	563.4 ± 0.26 ^α, a^	496.5 ± 0.22 ^α, a^	454.6 ± 0.21 ^α, a^	411.4 ± 0.20 ^α, a^
	95 °C, 5 min	Ca2.5	260.1 ± 0.40 ^b^	214.7 ± 0.27 ^b^	195.2 ± 0.26 ^b^	182.7 ± 0.11 ^b^	169.6 ± 0.24 ^b^
	105 °C, 5 min	Ca2.5	79.2 ± 0.20 ^c^	69.5 ± 0.14 ^c^	65.6 ± 0.12 ^c^	63.2 ± 0.06 ^c^	60.6 ± 0.08 ^c^
	Prior	Ca5	578.1 ± 1.16 ^α, a^	459.2 ± 0.39 ^α, a^	408.9 ± 0.27 ^α, a^	376.9 ± 0.20 ^α, a^	343.8 ± 0.18 ^α, a^
	95 °C, 5 min	Ca5	232.0 ± 0.14 ^b^	192.4 ± 0.06 ^b^	175.4 ± 0.05 ^b^	164.6 ± 0.05 ^b^	153.3 ± 0.05 ^b^
	105 °C, 5 min	Ca5	73.3 ± 73.3 ^c^	64.6 ± 0.11 ^c^	60.8 ± 0.08 ^c^	58.5 ± 0.06 ^c^	55.9 ± 0.03 ^c^
	Prior	Ca7.5	577.0 ± 0.86 ^α, a^	461.9 ± 0.31 ^α, a^	412.5 ± 0.17 ^α, a^	380.6 ± 0.16 ^α, a^	366.8 ± 0.16 ^α, a^
	95 °C, 5 min	Ca7.5	272.8 ± 0.23 ^b^	225.0 ± 0.16 ^b^	204.9 ± 0.14 ^b^	191.7 ± 0.13 ^b^	177.7 ± 0.12 ^b^
	105 °C, 5 min	Ca7.5	91.3 ± 0.18 ^c^	78.5 ± 0.11 ^c^	73.4 ± 0.08 ^c^	70.3 ± 0.05 ^c^	67.0 ± 0.04 ^c^
	Prior	Ca10	555.5 ± 0.88 ^α, a^	444.4 ± 0.30 ^α, a^	397.0 ± 0.25 ^α, a^	347.4 ± 0.11 ^α, a^	335.6 ± 0.10 ^α, a^
	95 °C, 5 min	Ca10	259.3 ± 0.31 ^b^	215.0 ± 0.25 ^b^	195.7 ± 0.23 ^b^	183.3 ± 0.19 ^b^	170.1 ± 0.11 ^b^
	105 °C, 5 min	Ca10	76.5 ± 0.17 ^c^	67.3 ± 0.10 ^c^	63.6 ± 0.08 ^c^	61.3 ± 0.06 ^c^	58.8 ± 0.05 ^c^
	Prior *	Ca40	482.0 ± 0.76	391.2 ± 0.43	348.9 ± 0.32	322.4 ± 0.33	295.3 ± 0.27
XG0.6AP1.4	Prior	Ca2.5	2129.3 ± 7.94 ^a^	1039.3 ± 2.83 ^a^	757.6 ± 1.26 ^a^	625.4 ± 1.27 ^a^	511.6 ± 1.02 ^a^
	95 °C, 5 min	Ca2.5	1958.4 ± 6.89 ^b^	934.2 ± 2.51 ^b^	674.8 ± 1.05 ^b^	554.7 ± 1.04 ^b^	452.2 ± 0.88 ^b^
	105 °C, 5 min	Ca2.5	1698.0 ± 4.90 ^c^	806.5 ± 1.82 ^c^	579.9 ± 0.84 ^c^	474.4 ± 0.87 ^c^	384.9 ± 0.72 ^c^
	Prior	Ca7.5	2245.7 ± 7.63 ^a^	1109.7 ± 3.49 ^a^	807.7 ± 1.54 ^a^	666.3 ± 1.49 ^a^	544.6 ± 1.29 ^a^
	95 °C, 5 min	Ca7.5	1883.6 ± 5.68 ^b^	930.0 ± 2.31 ^b^	676.5 ± 1.24 ^b^	557.8 ± 1.15 ^b^	455.8 ± 0.98 ^b^
	105 °C, 5 min	Ca7.5	1746.8 ± 5.19 ^c^	841.5 ± 2.13 ^c^	606.2 ± 1.11 ^c^	496.7 ± 1.10 ^c^	403.5 ± 0.98 ^c^

* Control without statistical comparison. ^†^ Means ± SD of triplicate analyses are given. ^a–c^ Means with different lowercase superscripts within an individual thermal treatment at a specific shear rate (sub-column) differ significantly. ^A–B^ Means with different lowercase superscripts indicate significant differences due to pH values for AP2.0 matrices; ^α–β^ Means due to Ca (CCA) concentrations for AP2.0 matrices; ^1–4^ Means due to composite ratio of formula.

**Table 5 foods-13-00090-t005:** Texture profile analysis (TPA) of the [XG0.6AP1.4-Ca7.5] of hydrocolloidal matrix prior to and after a 95 °C-5 min thermal processing.

Processing	Hardness (N/m^2^)	Adhesiveness (J/m^3^)	Cohesiveness	Gumminess(N/m^2^)
Prior	583.14 ± 25.60	19.74 ± 1.42	0.73 ± 0.03	423.84 ± 14.54
95 °C-5 min	487.65 ± 7.26 **	18.46 ± 2.66	0.70 ± 0.02 *	340.76 ± 13.27 **

n = 9 represent the standard deviations based on nine replicates, *, ** *t*-test for (*p* < 0.05) or (*p* < 0.01).

## Data Availability

Data is contained within the article or appendix material.

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
