# Peer review of "Effects of Calcium and pH on Rheological Thermal Resistance of Composite Xanthan Gum and High-Methoxyl Apple Pectin Matrices Featuring Dysphagia-Friendly Consistency"

_foods, 2023, doi:10.3390/foods13010090_

Round 1

Reviewer 1 Report

Comments and Suggestions for Authors

Manuscript Title: Effects of Calcium and pH on Rheological Thermal Resistance of Composite Xanthan Gum and High-methoxyl Apple Pectin Matrices Featuring Dysphagia-Friendly Consistency

The submitted manuscript presents an innovative approach to exploring rheological modifications in apple pectin and xanthan gum composites, aimed at developing dysphagia-friendly food products. The focus on characterizing food polysaccharides under various processing conditions is highly relevant and aligns well with the theme of the special issue on "Food Polysaccharides, Starch, and Protein: Processing, Characterization, and Health Benefits." The methodology employed is robust, and the results are presented logically and coherently, particularly in identifying the optimal conditions for achieving weak-gel-like behavior in dysphagia-friendly formulations. This work contributes significantly to the field and provides valuable insights for food science research.

 However, to further enhance the manuscript, I recommend the following minor revisions:

 1.     In Table 4, there appears to be a discrepancy between the unit of shear rate mentioned in the table header and the entries within the table. Please verify and ensure consistency in the units used.

 2.     The section between lines 382-396, which lists the measured numerical viscosity of the prepared matrices, could benefit from a more streamlined presentation. Instead of listing all values, summarizing key trends could improve readability and comprehension.

 3.     In terms of clarity in comparative analysis/discussion with existing literature on polysaccharide behavior in similar food matrices, some experimental conditions discussed require further elaboration for clarity. Expanding on how these conditions relate to and differ from the present study would offer a more comprehensive understanding of the research context.

 These minor adjustments will enhance the manuscript's clarity and overall impact. The reviewer recommends the manuscript for publication following these revisions, as it stands to make a valuable contribution to the understanding of polysaccharide behavior in food science, particularly in the context of creating dysphagia-friendly food products.

Author Response

As the attached PDF file.

Reviewer 2 Report

Comments and Suggestions for Authors

The article has some potential points to be addressed:

  1. The IDDSI framework, while emphasizing consistent viscosity for dysphagia-friendly matrices, does not include a direct measurement of viscosity. This might be considered a limitation as viscosity is a crucial parameter in determining the suitability of food for dysphagic individuals.

  2.  
  3. The article acknowledges challenges in accessing rheological testing equipment and expertise for more comprehensive studies. This limitation could affect the depth and scope of the research, potentially impacting the reliability of the findings.

  4.  
  5. Starch-based thickened matrices, commonly used in dysphagia-friendly formulations, face viscosity reduction due to amylase-containing saliva during oral processing. This could pose a risk of accidental inhalation and aspiration pneumonia for dysphagic individuals.

  6.  
  7. Starch-based thickened matrices may experience texture depletion during thermal processing, leading to a significant viscosity reduction. This finding could impact the stability and suitability of such matrices for dysphagia patients.

  8.  
  9. While exploring alternative thickeners, the article introduces pectin materials, which are chemically complex polysaccharides. The complexity of these materials may introduce challenges in terms of formulation, standardization, and patient acceptance, which could be considered a potential drawback.

Comments on the Quality of English Language

Moderate editing of English language required

Author Response

As the attached PDF file.

Round 2

Reviewer 2 Report

Comments and Suggestions for Authors

The responses from the authors address some concerns, but there are areas where the responses could be more thorough.

  1. Viscosity Measurement:

    • The reviewer rightly points out the absence of a direct measurement of viscosity in the IDDSI framework, considering it a limitation.
    • The authors respond by explaining the empirical nature of the framework and introducing additional standards for viscosity. However, the response does not directly address the concern about the absence of a direct viscosity measurement in the IDDSI framework. It would be beneficial to acknowledge this limitation explicitly and discuss how it might impact the study's findings and implications.
  2. Challenges in Rheological Testing:

    • The reviewer expresses concern about limitations in accessing rheological testing equipment and expertise, potentially affecting the research's depth and reliability.
    • The authors respond by justifying their motivation for accessing rheology but do not explicitly address the potential impact of these limitations on the reliability of their findings. A more detailed discussion of how these challenges were mitigated or how they might have influenced the study's outcomes would enhance the response.
  3. Risk of Accidental Inhalation:

    • The reviewer raises a valid point about the risk of accidental inhalation and aspiration pneumonia for dysphagic individuals using starch-based matrices.
    • The authors respond by agreeing with the concern and explaining the choice of non-starch-based matrices. However, the response lacks a clear acknowledgment of the potential risks associated with starch-based matrices and how the chosen matrices mitigate these risks. A more explicit discussion of the safety aspects would strengthen the response.
  4. Texture Depletion in Thermal Processing:

    • The reviewer points out the potential texture depletion in starch-based matrices during thermal processing and its impact on stability for dysphagia patients.
    • The authors respond by acknowledging the viscosity depletion and advocating for their non-starch-based formula. While this response addresses the concern, a more detailed discussion on the implications of viscosity reduction and how their formula addresses these challenges would enhance clarity.
  5. Complexity of Pectin Materials:

    • The reviewer expresses concern about the complexity of pectin materials and its potential challenges in formulation, standardization, and patient acceptance.
    • The authors respond by appreciating the comment and introducing information about the DE value of the pectin sample. While the response provides additional information, a more explicit discussion on how the complexity of pectin materials was managed in the study and its implications would improve the response.
  6. Moderate Editing of English Language:

    • The reviewer suggests moderate editing of the English language.
    • The authors respond by indicating that they have conducted the editing. Without specific examples or clarifications on the changes made, it's challenging to evaluate the effectiveness of this response.

In summary, while the authors have made efforts to address the reviewer's comments, more explicit and detailed discussions on the acknowledged limitations and potential risks associated with the study would strengthen the responses.

Comments on the Quality of English Language

 Moderate editing of English language required
